# MAT-PointPillars: Enhanced PointPillars algorithm based on multi-scale attention mechanisms and transformer

Xinpeng Yao[1], Peiyuan Liu[2]*, Jingmei Zhou[2]*, Zijian Wang[1,3], Songhua Fan[1], Yuchen Wang[2]

1 Shandong Key Laboratory of Smart Transportation (Preparation), Jinan, China, 2 School of Electronics and Control Engineering, Chang'an University, Xi'an, China, 3 School of Information Engineering, Chang'an University, Xi'an, China

* peiyuanliu@chd.edu.cn (PL); jmzhou@chd.edu.cn (JZ)

## Abstract

Aiming at the problem that small and irregular detection targets such as cyclists have low detection accuracy and inaccurate recognition by existing 3D target detection algorithms, MAT-PointPillars (Multi-scale Attention and Transformer PointPillars), a 3D object detection algorithm, extends PointPillars with multi-scale vision Transformers and attention mechanisms. First, the algorithm employs pillar coding for semantic point cloud encoding and introduces an attention mechanism to refine the backbone's upsampling process. Furthermore, the Transformer Encoder is introduced to improve the upsampling structure of the third stage of the backbone. On the KITTI dataset, our algorithm achieved 3D average detection accuracy (AP3D) of 81.15%, 62.02%, and 58.68% across three difficulty levels. Compared with the baseline model, the proposed algorithm improves AP3D by 2.44%, 1.19%, and 1.23% respectively. The real-time 3D object detection system is built based on ROS, and average running frames per second of the system is 22.63, which is higher than the sampling frequency of conventional LiDAR. By ensuring sufficient detection speed, the MAT-PointPillars algorithm can increase detection accuracy of cyclists in real-world scenarios.

## 1. Introduction

In the realm of autonomous driving, three-dimensional(3D) object detection is a critical component for environmental perception, enabling precise identification and localization of objects. This capability is fundamental for path planning and decision-making processes within the vehicle's operational environment. Despite the advancements in image-based recognition algorithms, they are often impeded by variable environmental factors such as lighting conditions and weather, particularly during nighttime, rain, and fog. In contrast, LiDAR-based systems offer a more reliable

**Data availability statement:** The data that support the findings of this study are openly available in KITTI at https://www.cvlibs.net/datasets/kitti/eval_object.php?obj_benchmark=3d reference number [30]. We used the datasets: left color images of object data set, Velodyne point clouds, camera calibration matrices of object data set, training labels of object data set.

**Funding:** This work was supported in part by Open Project of Shandong Key Laboratory of Smart Transportation (Preparation) under Grant 2021SDKLST004, National Natural Science Foundation of China under Grants 52472337 and 52302491, China Postdoctoral Science Foundation under Grant 2023T160129, Key Science and Technology Project of Ministry of Transport under Grant 2022-ZD6-079, Research Funds for the Interdisciplinary Projects, CHU under Grant 300104240911, Natural Science Basic Research Program of Shaanxi under Grant 2023-JC-YB-523, Key Research and Development Program of Shaanxi under Grant 2023-YBGY-119. The main research work of this manuscript is 3D object detection. The funders have provided financial support for the key research of 3D object detection algorithms for autonomous driving and have also provided a solid theoretical basis, contributing to the successful completion of various stages of the study. Their assistance helped enable the investigation and advancement of this work.

**Competing interests:** The authors have declared that no competing interests exist.

approach by providing accurate 3D coordinates of objects, unaffected by such conditions [1].

Early 3D object detection algorithms essentially directly processed the original point cloud. Yang et al. [2] proposed a farthest point sampling method to retain more foreground point information in the sampled point cloud, combined spatial distance and feature distance, and designed a candidate point generation module for feature fusion. Because of the large amount of data in the point cloud, encoding and decoding the point cloud directly takes a great deal of time, which hampers the ability to meet the needs of real-time detection. Therefore, the disordered point cloud is divided into regular voxels, so that the point clustering in each voxel is the global feature, and 3D convolution is used to extract the feature and regress the category and position of the target. In the process of sampling voxel features under the hierarchy, the resolution drops, which makes the location edge of the target frame fuzzy. Shi et al. [3] proposed a two-stage 3D object detection algorithm PV-RCNN, which extracts the original features of the point cloud and then revises the target frame in the second stage. The accuracy and speed of 3D object detection are balanced while preserving the original features of point cloud. Xu et al [4] proposed bi-stream generative model to learn the fine-grained representations fused with camera-invariant global feature and pedestrian-aligned local feature to solve Camera-Camera problem and the Camera-Person problem. Tao et al [5] proposed the Enhanced feature extraction-You Only Look Once (EFE-YOLO) algorithm to improve the detection of industrial small objects. Sun et al [6] proposed an end-to-end multi-scale residual network with parallel attention mechanism to address the problem of fault diagnosis contains noise disturbances, small samples, compound faults, and mixed conditions.

Based on the above analysis, this paper proposes a 3D object detection algorithm that integrates a multi-scale attention module and Transformer Encoder to improve the PointPillars algorithm. The specific work is as follows:

(1) The Attention module is introduced in the upsampling modules of the first and second stages at the backbone network, and the Attention Deconv module is constructed to increase the ability of the network to extract multi-scale features.

(2) The Transformer Encoder is introduced to improve the third-stage upsampling module of the Transformer backbone network and build Transformer Deconv to increase the network's ability to both extract small-scale features and capture features of non-rigid targets.

(3) Model training and method validation is conducted on the KITTI 3D Object dataset to evaluate the accuracy of the proposed algorithm, and ablation experiments are conducted on different attention and Transformer layers, and the head numbers of Self-attention modules, to verify the effectiveness of the improved module.

(4) A real-time 3D object detection system is built based on ROS (Robot Operating System). The Tanway Scope 64 LiDAR is used to collect LiDAR data packets on a real-world road surface to verify the generalization and real-time detection ability of the proposed algorithm.

This paper is structured as follows: The second section introduces the relevant methods of 3D object detection algorithm. The third section describes the construction of the proposed MAT-PointPillars algorithm. The experimental results are presented in Section 4. Finally, the fifth section summarizes the work of this paper and proposes future areas of research.

## 2. Related work

At present, the 3D object detection algorithms are mainly divided into three categories: point-based object detection, voxel-based object detection and multi-view projection-based object detection.

The detection method based on point cloud mainly focuses on the local characteristics of each point. A point cloud has a large amount of data that is sparse and disordered. Qi et al. [7,8]proposed the PointNet/PointNet++ network, which provides the basic backbone network of point-based detection methods. The network uses the full connection layer to map the point cloud information to the high-dimensional space, and the pooling layer fuses the point cloud features. Using a coding-decoding structure, the feature resolution of point cloud feature extraction can be maintained. Shi et al. [9] proposed a two-stage object detection algorithm, PointRCNN, which decomposes detection tasks into instance segmentation and regression tasks. In the first stage, candidate regions are generated based on foreground points, and redundant target frames are removed by the Non-Maximum Suppression (NMS) algorithm. In the second stage, feature fusion is performed and the target frame is modified. The algorithm strengthens the fine regression of the target frame and improves the detection accuracy. Yang et al. [10] proposed a new two-stage 3D object detection framework, named sparse-to-dense 3D Object Detector (STD). However, these algorithms are computationally intensive and inefficient.

The PointPillars based on voxel detection algorithm further reduces the calculation amount by using cylinder voxelization and two-dimensional (2D) convolution, and has the advantages of lightweight algorithm structure design. Zhou et al. [11] put forward the VoxelNet algorithm, which divides the point cloud into 3D voxels, introduces a feature coding layer to transform the points in each voxel into the same feature representation, and finally generates detection results using Region Proposal Network (RPN). The algorithm yields high feature extraction efficiency and realizes end-to-end detection. However, there are still some disadvantages, such as a large number of calculation parameters, slow inference speed, and low direction estimation performance. Voxel-RCNN [12] uses a combination of 2D and 3D convolution, but the network complexity is greatly increased. Yan et al. [13] proposed the SECOND algorithm. Because the original point cloud data has a sparse data structure, a sparse 3D convolutional neural network is used to improve the inference speed. SECOND introduces the use of a new data enhancement method and angle loss regression form to improve the performance of direction estimation and convergence speed. PillarNeXt [14] uses Dilated Convolution to improve the accuracy of the vehicle, but the improvement of small targets is still not obvious. Lang et al. [15] proposed the PointPillars object detection method, which voxelizes the point cloud in the form of pillar. The pillar reduces the 3D point cloud space to a 2D BEV and constructs pseudo-image features, which supports use of 2D convolutional neural networks for end-to-end point cloud feature learning and object detection. Yang et al. [16] improved on the removal of input point cloud noise, but lacked thinking about the target point cloud.

The multi-view projection method projects 3D point cloud data onto a 2D plane, and uses an image-based 2D object detection model to process the projection data. The method mainly adopts two projection modes: front view (FV) and bird's eye view (BEV). FV projection can not only convert sparse discrete point clouds into denser projection images, but also fuse with color images captured by vehicle cameras to provide richer semantic information for object detection. However, FV projection can easily cause occlusion between target objects. Relatively speaking, BEV projection can preserve the spatial distribution of the target in the point cloud, avoid the occlusion problem, and basically maintain the physical size of the target object, which is very beneficial for the regression analysis of the target position. Cheng et al. [17] proposed MV3D, a 3D object detection algorithm based on multiple viewing perspectives, which projects LiDAR point clouds into FV and BEV; constructs a 3D RPN to extract features from FV, BEV, and color images; and generates

a target candidate region, thus realizing feature fusion and target localization through a regional fusion network. This method achieves more accurate object detection by means of multi-view projection and multi-source feature fusion, but the complex feature fusion process leads to a long inference time, which cannot meet the real-time processing requirements. Ku et al. [18] proposed the AVOD algorithm to project the original point cloud from the BEV perspective, to form a multi-channel BEV image. Liang et al. [19] proposed an end-to-end learnable architecture that can explain 2D and 3D object detection as well as ground estimation and depth completion. All of these tasks are complementary, helping the network learn better by fusing different levels of information. Liang et al. [20] proposed a new type of 3D object detector that can use LiDAR and cameras for very precise positioning. An end-to-end learnable architecture is designed to use continuous convolution to fuse images and LiDAR feature maps of different resolution levels. Qi et al. [21] utilizes mature 2D object detectors and advanced 3D deep learning for object localization, achieving efficiency and high recall rates even for small objects. By learning directly in the original point cloud, the algorithm is also able to accurately estimate 3D bounding boxes, even under strongly occluded or very sparse points. Xu et al. [22] proposes a general 3D object detection method that utilizes image and 3D point cloud information. Image data and raw point cloud data are processed independently by the CNN and PointNet architectures, respectively. The resulting output is then combined by a new fusion network that uses the input 3D points as spatial anchors to predict multiple 3D box hypotheses and their confidence levels.

In recent years, PointPillars has been improved in several areas. The Frustum-PointPillars [23] algorithm adopts multimodal input and uses 2D image information to propose a cone prior area, and then voxelizes the point cloud in the cone area by joint calibration of camera and LiDAR. Although this approach improves detection accuracy, it relies heavily on the detection results provided by the 2D detector. When the accuracy of the 2D detector is affected, the accuracy of the 3D object detection will also be greatly affected, which will in turn impact the calibration accuracy, making the algorithm's results susceptible to the influence of external factors. The color-PointPillars [24] method, for a sparse color point cloud, inputs the calibrated camera picture and point cloud at the same time, and makes a one-to-one match between the LiDAR and image pixels through joint calibration, so that the point cloud can obtain the RGB color of the target and enhance the ability of the network to capture multi-scale spatial features of the point cloud. This method relies on the accuracy of the joint calibration parameters, so there will be calibration errors that affect detection. The Attention-PointPillars [25] method, based on the Attention module, strengthens the feature extraction capability of pseudo-images by introducing spatial attention and channel attention, so as to enhance the feature modeling capability of the algorithm, while ignoring the global features of point cloud. Chen et al. [26] proposed a LiDAR 3D object detection method based on improved PointPillars, and used the Swin Transformer [27] to improve the 2D convolution downsampling module of PointPillars. It enables the network feature extraction stage to use self-attention to enrich context semantics and obtain global features, enhances the ability of the network to capture multi-scale spatial features of the algorithm, and improves the angle-of-sight accuracy of the algorithm, but does not solve the problem of low detection accuracy for small targets.

In order to solve the problems in the existing methods, this study proposes an improved 3D object detection algorithm, MAT-PointPillars, which integrates multi-scale attention mechanism and transformer encoder. Based on PointPillars, the algorithm introduces the Attention module to improve the first and second stages' upsampling module of the backbone network, which enhance the feature extraction capability of the network in large-scale feature maps; Transformer Encoder is introduced to improve the upsampling structure of the third stage of the backbone network, enrich the semantic information of sparse point clouds of small targets, enhance the feature extraction capability of the network for small-scale targets, and improve the detection accuracy.

## 3. Enhanced PointPillars algorithm based on multi-scale attention mechanisms and transformer

### 3.1. The overall framework of proposed algorithm

The overall network architecture designed in this paper incorporates the multi-scale attention module and the Transformer 3D object detection algorithm, as shown in Fig 1. The algorithm inputs point cloud data and outputs the 3D enclosing



**Fig 1. MAT-PointPillars network structure diagram.** The network is composed of a cylindrical coding module, backbone network, and detection head. The cylindrical coding module first divides the 3D space into equal cylindrical regions, and then stacks them to form false images. The backbone network is composed of a basic convolutional block, an upsampling module based on a multi-scale attention module, and a small-scale feature extraction module based on Transformer.

frame coordinates, class information, and confidence of the target. The network is composed of a cylindrical coding module, backbone network, and detection head. In the cylindrical coding module, first the input data consists of points in 3D space collected by LiDAR. The point cloud is transformed into a columnar structure, with each column representing a region in the point cloud. The features in the columnar structure are extracted through the learning process. The results of feature learning are stacked into pseudo picture for easy subsequent processing. The backbone network is composed of a basic convolutional block, an upsampling module based on a multi-scale attention module, and a small-scale feature extraction module based on Transformer. First, initial convolution operation is performed to extract basic features from input data. The figure shows three branches followed by a deconvolution layer. This design allows the network to extract features from different scales and angles. Feature maps from different sources are spliced together in channel dimension to fuse multi-scale information. Finally, the outputs of all branches are combined through concatenation operations to form an output that integrates multi-scale features. Among them, the upsampling module based on a multi-scale attention module fuses the Convolutional Block Attention Module (CBAM) and Coordinate Attention (CA) at different scales to enhance the network extraction of spatial and channel dimension feature information. The small-scale feature extraction module

based on Transformer integrates the Transformer Encoder and the upsampling module to enhance the network's ability to extract small-scale features.

## 3.2. Pillar coding module

In MAT-PointPillars algorithm, pillar coding is adopted for point cloud data in the coding process. The original point cloud data in 3D space is $(x, y, z, r)$, where $(x, y, z)$ represents the coordinates of the point cloud in 3D space and $r$ represents the reflection intensity of the point cloud. Taking KITTI's LiDAR coordinate system as the reference coordinate system, the 3D space is divided into cubic pillars of the same volume under this coordinate system, and all point clouds in the 3D space are allocated to cubic pillars of equal size, as shown in Fig 2.

The point cloud data within each column is enhanced and offset information is added to each point. The enhanced point is represented as $(x, y, z, x_c, y_c, z_c, x_p, y_p)$, in which $(x_c, y_c, z_c)$ represents the offset between the point and the average coordinate value of all point clouds in the pillar, and $(x_p, y_p)$ represents the offset from the center coordinate of the pillar. $z_p$ is added in the implementation of this algorithm and indicates the offset from the center of the z axis in the pillar where the point is located. After the data is enhanced, it is expressed as $(x, y, z, x_c, y_c, z_c, x_p, y_p, z_p)$. After this, the point cloud data is represented as $(D, P, N)$, where $D$ represents the characteristic latitude of the point cloud, $P$ represents a non-empty pillar, and $N$ represents the number of point clouds in each pillar. The data of negative axis $x$ is intercepted. At the same time, points that are too far away are considered too sparse. To ensure the reliability of detection, the selection range of point cloud space is shown in Table 1.

The maximum number of point clouds in each pillar is set to $N_{thred}$ points. If there are more than $N_{thred}$ points, random sampling is performed. If there are less than $N_{thred}$ points in a pillar, zero samples are used to fill the pillar. After data enhancement, the data is represented as $(D, P, N)$,. The PointNet network is used to extract features from the point cloud data, and the obtained data is represented as $(C, P, N)$. After the maximum pooling output is $(C, P)$, the obtained data is returned to the corresponding position to form a pseudo-image.

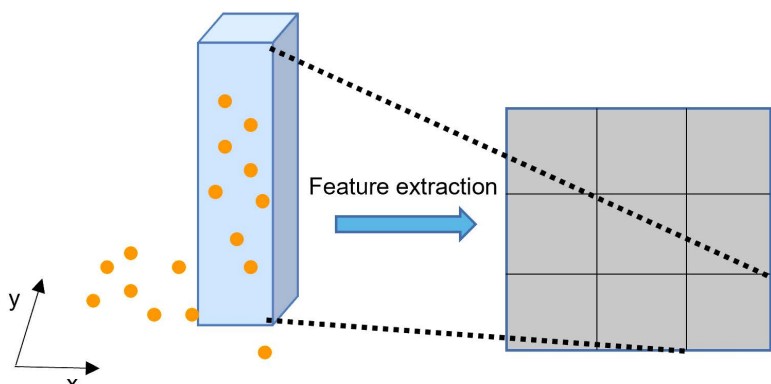

**Fig 2. Schematic diagram of the pillar encoding.** Pillar coding is a technique for processing point cloud data or 3D spatial data by collecting points in 3D space to simplify data processing or for further analysis.

**Table 1. Range of point clouds.**

| Point cloud extent | x | y | z |
|---|---|---|---|
| Start coordinates | 0 | −39.68 | −3 |
| End coordinates | 69.12 | 39.68 | 1 |

## 3.3. Upsampling module based on multi-scale attention

The algorithm optimizes the upsampling process of the first and second stages by integrating attention modules. The optimized upsampling module includes the attention mechanism layer and the upsampling layer. The improved upsampling module integrated with CBAM attention module [28] consists of three sub-modules, the channel attention module (CAM), the spatial attention module (SAM), and the upsampling module, which extract attention features from two dimensions of channel and space respectively, as shown in Fig 3. For CAM, the channel dimension is unchanged, the spatial dimension is compressed, and the content information of the input feature map is the focus. The input feature map is passed through two parallel Maxpooling layers and Averagepooling layers, and the feature map is changed from $C \times H \times W$ to $C \times 1 \times 1$. Then it is passed through the Share MLP module, in which the number of channels is first compressed to $1/r$ (Reduction) times. Then expand to the original number of channels, and get two activation results after ReLU activation function. These two outputs are added element-by-element, and then the output of Channel Attention is obtained through a sigmoid activation function. For SAM, the spatial dimension is unchanged, the channel dimension is compressed, and the target position information in the feature map is the focus. Channel Attention output results are obtained by maximum pooling and average pooling to obtain two $1 \times H \times W$ feature graphs, then Conv operation is used to concatenate the two feature graphs, and then a sigmoid is used to obtain Spatial Attention feature graphs. Through the combination of CAM and SAM, the network sufficiently accommodates the characteristics of the two dimensions of space and channel, and improves classification and accuracy.

The improved upsampling module of fused CA module [29] is shown in Fig 4, which considers not only channel information, but also location information related to direction. The global average pooling along the width and height of each channel is performed to obtain the feature map, capture the information in the direction of height and width, splicing the two pooled feature maps on the channel dimension, and using the two-dimensional convolution layer (Conv2d) to fuse

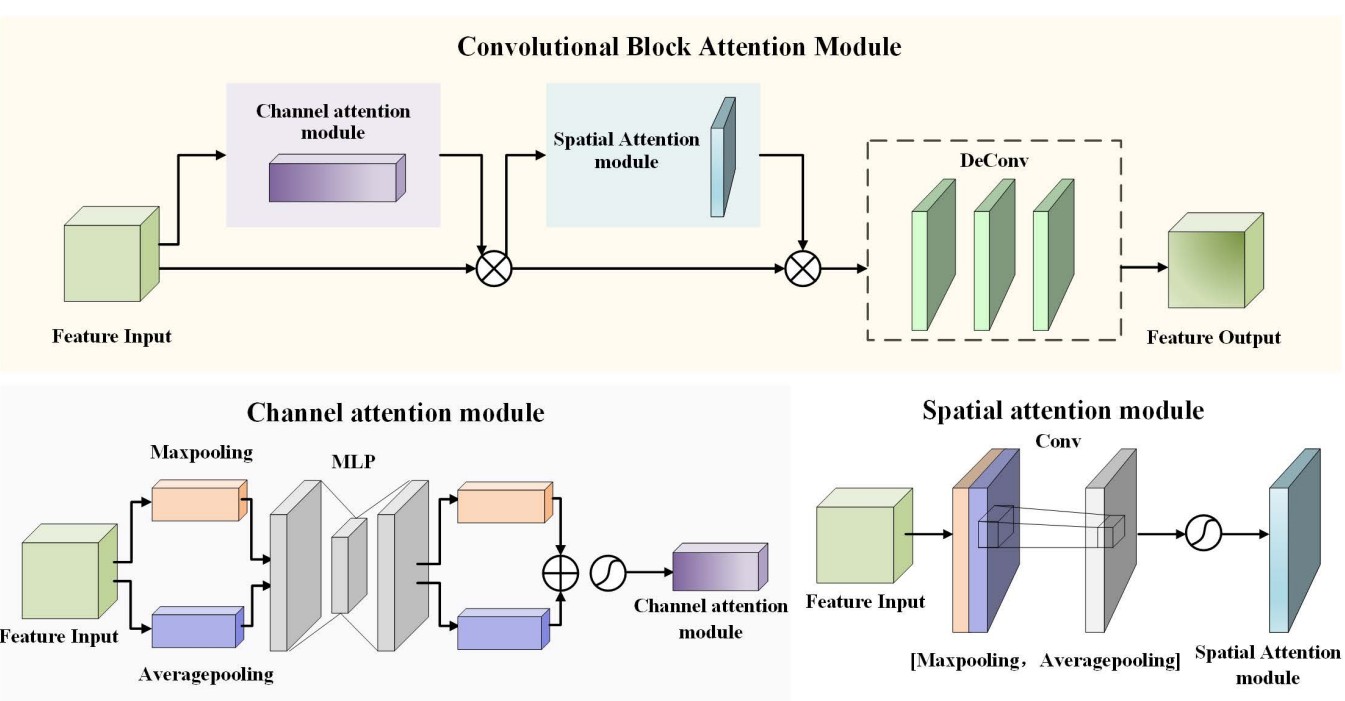

**Fig 3. Improved upsampling module incorporating the CBAM attention module.**

and transform the features. The convolutional feature maps are normalized in batches (BatchNorm), and the expressiveness of the model is increased by nonlinear activation function. The feature maps after batch normalization and activation are split into two feature maps (Split). The feature maps after splitting are processed separately by two other two-dimensional convolution layers. The Sigmoid activation function is applied to the two convolution feature maps to generate two attention graphs, which will recalibrate the input feature maps in width and height respectively. The active attention map is multiplied by the original input feature map to reweight the original features and to strengthen or weaken certain features. Output the weighted feature map, which combines the channel information and spatial position information of the original input feature.

### 3.4. Feature extraction module based on transformer

The proposed algorithm transforms the point cloud into pseudo-images through pillar feature extraction to meet the input of the Transformer [30] network structure. The input of the algorithm is the input sequence after flattening the pseudo-images. The input data format is $(H, W, C)$, and it is divided into $N$ $L \times L$-sized image blocks and then flattened to obtain the input sequence, where $H$, $W$, and $C$ are, respectively, the height of the images and the width and number of channels.

In order to improve the ability to extract features of small targets, Zhu et al. [31] proposed the TPH-YOLOv5 algorithm, which replaces the CSP module with a Transformer Encoder, thus greatly improving the detection accuracy of small targets. Inspired by this, this paper uses the Transformer coding layer to improve the upper sampling layer of the third stage of the backbone network and enhance the ability of the network to extract small targets. The architecture of the Transformer Encoder is shown in Fig 5, where the Multi-Head Attention component consists of multiple Self-attention modules, The self-attention mechanism allows the model to process each word while taking into account the relationships between all other words in the input sequence. The Multi-Head Attention mechanism allows the model to learn different semantic information in different representation subspaces by making multiple linear transformations of the input and calculating multiple sets of attention scores. Norm is a normalized operation. Normalization can ensure that there is a similar range of values between different features or variables, helping to improve the performance and convergence speed of the model. And MLP is a multi-layer perceptron. The network uses Droppath instead of the traditional Dropout, randomly ignoring certain sublayers during training and using full numbers when in use.

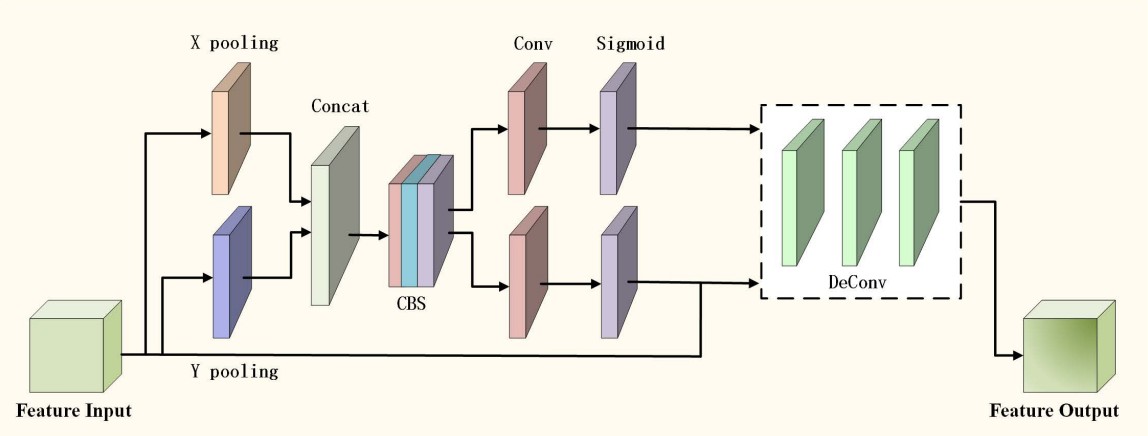

**Fig 4. Improved upsampling module incorporating CA module.** Firstly, the global average pooling is decomposed and pooled according to x dimension and y dimension respectively. After splicing, the CBS module is used to extract features. Finally, the output features are spliced and convolved, and then the feature map is up-sampled to make full use of spatial dimension information and strengthen the ability of the network to extract spatial features.

# Transformer Encoder

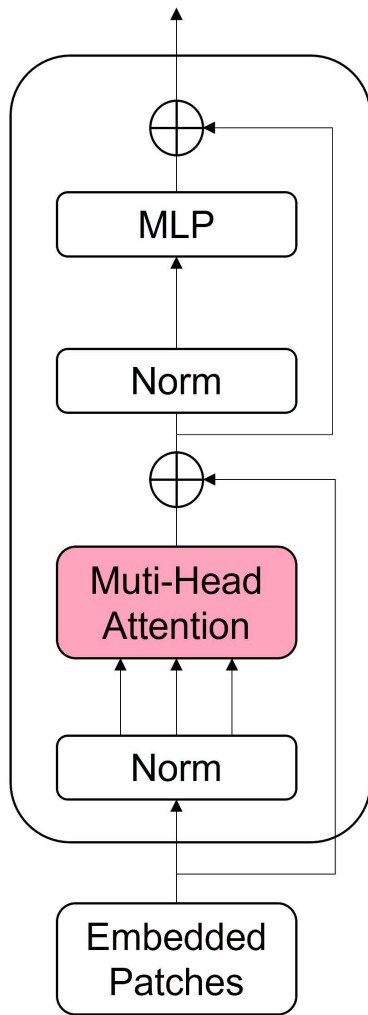

**Fig 5. Transformer Encoder network structure.**

Transformer Encoder can capture global information and rich context information, leveraging the Self-attention module to tap into the feature characterization potential. The network structure diagram of the small-scale feature extraction module based on Transformer [30] is shown in Fig 6. Input features are first passed through the transformer encoder. Then the features processed by the deconvolution layer are output. This process can enhance the ability of the network to capture multi-scale spatial features. The improved upper sampling layer consists of a Transformer Encoder module and a Transformer upsampling module.

## 4. Experimental results and analysis

### 4.1. Environment and dataset

The network model is built based on the PyTorch framework and deployed on Intel® Core™ i7-12700 and RTX3060ti for network structure training and result verification. Open3D [32] and Mayavi are used to visualize the point cloud detection results. This experiment processed two samples per batch, with a total of 150 iterations.

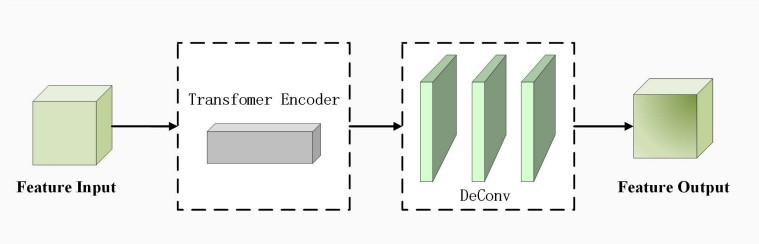

**Fig 6. Improved upsampling module based on Transformer.**

The KITTI dataset [33] is used to evaluate and verify the algorithm. The proposed algorithm is evaluated using primarily the LiDAR point cloud data in the data set, including images collected in urban areas, rural areas, highways, and other scenarios, with multi-source data sampling and synchronization conducted at a frequency of 10 hertz. According to the degree of target occlusion and truncation, the scene objects are divided into three difficulty levels: easy, moderate, and hard. Table 2 shows the meanings and data examples of each field in the data set label file.

## 4.2. Evaluation criteria

In this experiment, IoU3D and IoUBEV are used to evaluate whether the target frame is matched with the true value of the hit. IoU3D refers to the intersection ratio between the detection results and the true value in 3D space, and IoUBEV refers to the intersection ratio between the detection results and the true value in the aerial view. If the Intersection Over Union (IoU) is greater than the threshold, the detection is considered successful; otherwise, the detection may be erroneous. The higher the IoU threshold is, the stricter the judgment standard is. For cyclists, there are two IoU thresholds for KITTI dataset evaluation indicators, 0.5 and 0.25, and the IoU threshold for this experiment is 0.5.

For the above hit judgment method, accuracy and recall rate can be calculated, as shown in Equations (1) and (2):

$$Precision = TP/(TP + FP) \tag{1}$$

$$Recall = TP/(TP + FN) \tag{2}$$

**Table 2. Examples of truth label file formats for the KITTI dataset.**

| Name | Description | Unit |
|---|---|---|
| Type | The KITTI dataset includes a variety of object categories, such as vehicles, pedestrians, and cyclists. | – |
| Truncated | The degree of target truncation is the floating-point number in the interval [0,1], and KITTI indicates the degree of truncation by calculating the distance between the target and the image boundary. | – |
| Occluded | The target occlusion degree is an integer in the interval [0,3], where 0 is fully visible, 1 is partly occluded, 2 is mostly occluded, and 3 is fully occluded. | – |
| Alpha | The target viewing angle, which is the floating-point number in the $[-\pi, \pi]$ range. | radian |
| 2Dbbox | The 2D target box, which is the coordinates of the upper and lower left and right corners of the image. For example: (387.63, 181.54, 423.81, 203.12). | pixel |
| Dimensions | The dimensions of the 3D target box, i.e., height, width, and length. For example: (1.67, 1.87, 3.69). | m |
| Location | The position of the center point of the 3D target in the camera coordinate system, i.e., x-axis coordinates, y-axis coordinates, and z-axis coordinates. For example: (−16.53, 2.39, 58.49). In point cloud target detection, the transformation matrix stored in the sensor calibration file needs to be used to convert the position of the target center point to the LiDAR coordinate system. | m |
| Rotation | The orientation angle of the 3D target, which is the floating-point number in the $[-\pi, \pi]$ interval. | radian |



Where, *TP* indicates the number of detection boxes that are normally matched, *FP* indicates the number of detection boxes that are not matched or repeatedly checked, and *FN* indicates the model incorrectly predicts the instances of a positive class as a negative class. The AP (average precision) calculation in this experiment used two strategies: R11 and R40. R11 refers to the estimation of P-R curve area by the 11-point interpolation method, and R40 refers to the estimation of P-R curve area by the 40-point interpolation method, both of which are official metrics provided by KITTI dataset.

### 4.3. Overall algorithm performance analysis

After 150 iterations, the model tends to converge, and the total loss curve and the three-component loss curve tend to be smooth. Fig 7 shows the decline curve of the total loss and the three-component loss with the time step.

The average detection accuracy is used to conduct quantitative evaluation of the MAT-PointPillars algorithm in the KITTI validation set, and the results are shown in Table 3. When the R11 measurement is used, the proposed algorithm obtains 81.15%, 62.02%, and 58.68% of the 3D average detection accuracy AP3D and 83.85%, 66.38%, and 62.63% of the BEV target average detection accuracy APBEV for the cyclist class at the easy, moderate, and hard levels, respectively. When using the R40 metric, the paper obtained 83.03%, 62.51%, and 58.07% of the 3D average detection accuracy AP3D and 87.24%, 66.95%, and 62.46% of the BEV average detection accuracy APBEV for the cyclist class at the easy, moderate, and hard levels, respectively. These results show that the proposed algorithm can effectively identify objects in 3D space, and improve the network's recognition accuracy for cyclists.

To evaluate the detection performance of the algorithm, Open3D is used to visualize the detection results of the algorithm, as shown in Fig 8, in which the dark blue box is the real target box, the green box is the vehicle detection box, the yellow box is the cyclist detection box, and the light blue is the pedestrian detection box. Point cloud images of eight scenes are randomly selected in the data set for detection. The algorithm correctly identifies vehicles, pedestrians, and cyclists. Except in Fig 8e, f, there are cases of false detection and missing detection due to occlusion and the sparse number of point clouds. The visualization results show that the algorithm has good detection performance.

Table 4 compares the average detection accuracy of the proposed method with other methods. R11 is used for as the basis of measurement; that is, 11 interpolation points are used in AP calculation. The average detection accuracy of the model is the experimental result after iteration with the same hardware environment until the model converges.

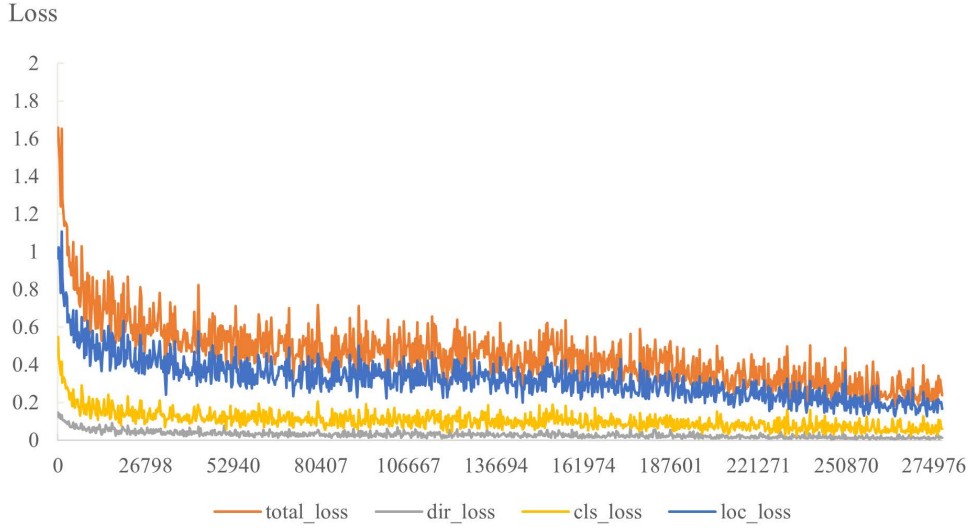

**Fig 7. The total loss and three-component loss.**

**Table 3. Quantitative evaluation results of the proposed 3D object detection algorithm on the KITTI dataset.**

| The proposed 3D object detection algorithm | Easy | Moderate | Hard |
|---|---|---|---|
| AP3D _R11(%) | 81.15 | 62.02 | 58.68 |
| APBEV _R11(%) | 83.85 | 66.38 | 62.63 |
| AP3D _R40(%) | 83.03 | 66.38 | 58.07 |
| APBEV _R40(%) | 87.24 | 66.95 | 62.46 |

As Table 4 indicates, the proposed MAT-PointPillars algorithm achieves an AP3D_R11(%) of 81.15%, 62.02%, and 58.68% on easy, moderate, and hard samples, which are higher than the benchmark model by 2.44%, 1.19%, and 1.23% respectively. The algorithm obtains 83.85%, 66.38%, and 62.63% of easy, moderate, and hard samples in the BEV, which is higher than the benchmark model 0.81%, 0.87%, and 0.15%, respectively. As Table 4 indicates, the proposed MAT-PointPillars algorithm achieves an AP3D_R11(%) of 81.15%, 62.02%, and 58.68% on easy, moderate, and hard samples, which are higher than the benchmark model by 2.44%, 1.19%, and 1.23% respectively. The algorithm obtains 83.85%, 66.38%, and 62.63% of easy, moderate, and hard samples in the BEV, which are higher than the benchmark model 0.81%, 0.87%, and 0.15%, respectively. Compared with VoxelNet, PillarNet, the AP3D_R11(%) and APBEV_R11(%) of MAT-PointPillars are higher than those of VoxelNet and PillarNet in easy, moderate, and hard tasks. Compared with SECOND, AP3D_R11(%) in easy and hard samples are higher than SECOND by 4.81%, 0.18% respectively. Our

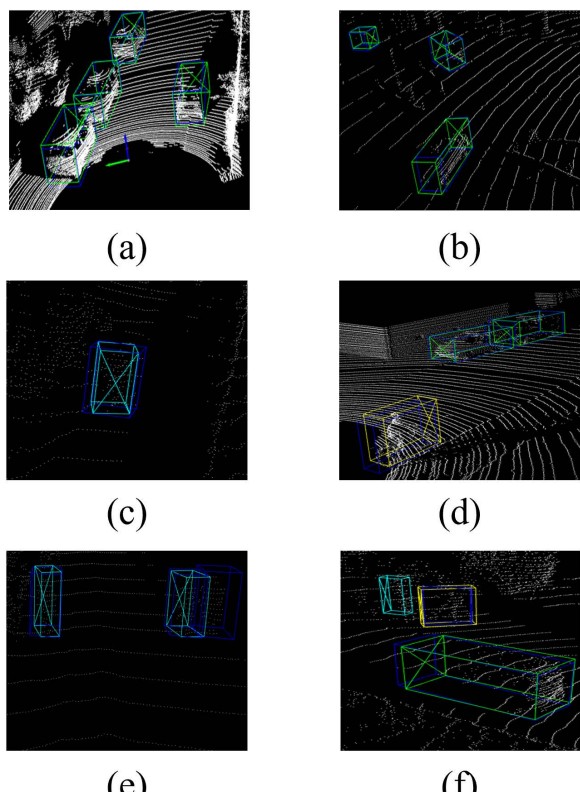

(a)  (b)

(c)  (d)

(e)  (f)

**Fig 8. Visualization of algorithm detection results.**

**Table 4. Comparison of average object detection accuracy between the proposed method and other methods in the KITTI dataset.**

| Method | Input | AP3D _R11(%) | | | APBEV _R11(%) | | |
|---|---|---|---|---|---|---|---|
| | | Easy | Moderate | Hard | Easy | Moderate | Hard |
| VoxelNet | LiDAR | 67.17 | 47.65 | 45.11 | 74.41 | 52.18 | 50.49 |
| PillarNet | LiDAR | 77.41 | 58.09 | 55.69 | 80.85 | 62.63 | 59.19 |
| SECOND | LiDAR | 76.34 | 62.20 | 58.50 | 81.62 | 68.58 | 63.65 |
| PointPillars | LiDAR | 78.71 | 60.83 | 57.45 | 83.04 | 65.51 | 62.48 |
| MAT-PointPillars | LiDAR | 81.15 | 62.02 | 58.68 | 83.85 | 66.38 | 62.63 |

algorithm mainly focuses on 3D object detection, so AP3D_R11(%) has good indicators. The algorithm achieves superior detection performance.

## 4.4. Ablation experiment

In this algorithm, Transformer Encoder is introduced in the third stage of upsampling. Ablation experiments are conducted on the head number of multiple-head attention in Transformer and the number of layers of Transformer Encoder, and upsampling modules with different attention are introduced. Including CBAM and CA, the accuracy of the experimental results is compared, as shown in Table 5. In the ablation experiment, we not only consider the changes in the number of model layers, but also conduct separate ablation experiments on key submodules such as CBAM and CA. For example, we remove the CBAM and CA modules, respectively, and observe the changes in model performance. Ablation results show that when CBAM is the type of attention module, increasing the number of Multi-head Attention modules alone will not increase the accuracy of model detection. However, the proposed algorithm has the best effect when increasing the number of layers and the number of Multi-head Attention modules in Transformer at the same time. In our experiments, the biggest number of Transformer Encoder layers is set to 2, as the number of Transformer Encoder layers increases, the computational complexity of the model also increases significantly, leading to longer inference times and negatively impacting real-time performance. Experiments show that the balance between accuracy and efficiency of 2-layer Transformer Encoder is the optimal solution under current hardware conditions. According to the results of ablation experiments, CBAM is superior to CA in detection accuracy, as shown in Table 5. Specifically, CBAM outperformed CA on both AP3D and APBEV. For example, the combination of CBAM+CBAM reached 81.15% on AP3D, while the combination of CA+CA is only 73.47%

## 4.5. Efficiency experiment

The hardware environment of this experiment included the Intel(R) Core (TM) i7-12700 and RTX3060ti. The system is ubuntu18.04 and the ROS version is Melodic. The Tanway Scope 64 LiDAR is used and the actual road scene is taken as the acquisition environment. A one-minute Rosbag is collected by LiDAR on the actual road at a frequency of 10 hertz. The detection interface of the real-time detection system of the simulated LiDAR is shown in Fig 9. The recorded bag is collected on real roads, and the green box is the detected traffic target.

**Table 5. Results of ablation experiment.**

| Attention | Batch | Head | Layer | AP3D | APBEV |
|---|---|---|---|---|---|
| CBAM+CBAM | 2 | 2 | 1 | 78.40% | 81.07% |
| CBAM+CBAM | 2 | 4 | 1 | 76.41% | 79.93% |
| CA+CA | 2 | 4 | 2 | 73.47% | 78.91% |
| CA+CBAM | 2 | 4 | 2 | 74.98% | 78.00% |
| CBAM+CBAM | 2 | 4 | 2 | 81.15% | 83.85% |

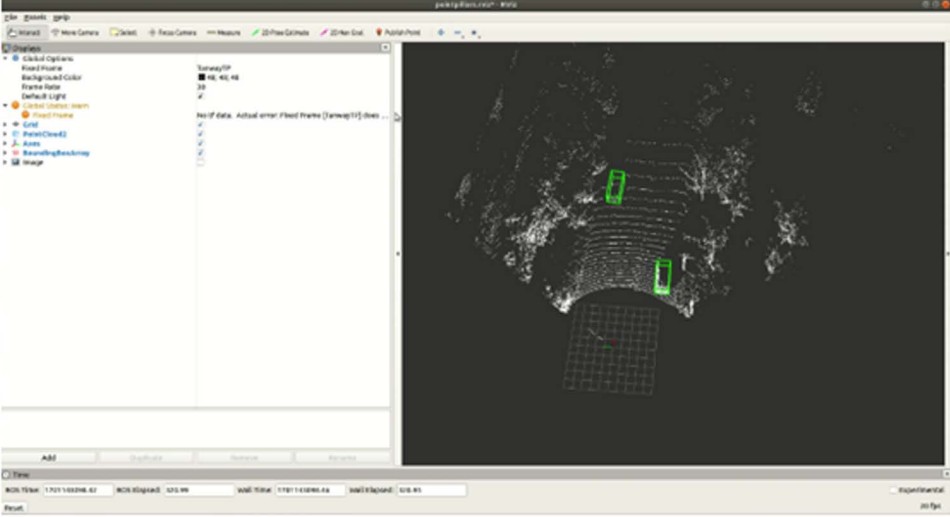

**Fig 9. Operation diagram of 3D real-time detection system.**

In the real-time detection system, 25 time-steps are randomly sampled, and the average display frame rate of the algorithm is 22.63 frames per second (fps). Systems running the PointPillars network display an average frame rate of 22.13 fps. Fig 10 shows the line chart of frame rate displayed by two algorithms for real-time 3D object detection. As Fig 10 indicates, the real-time detection system displays a frame rate between 17 and 25 fps during operation, which meets the requirements of real-time detection and verifies the real-time performance of the proposed algorithm.

## 5. Conclusions

Accurate and real-time detection of 3D objects plays a crucial role in an automated driving system's ability to make decisions and safely avoid obstacles. Therefore, this paper proposes a real-time 3D object detection algorithm, termed MAT-PointPillars, that integrates a multi-scale attention module and the Transformer Encoder. Based on the PointPillars

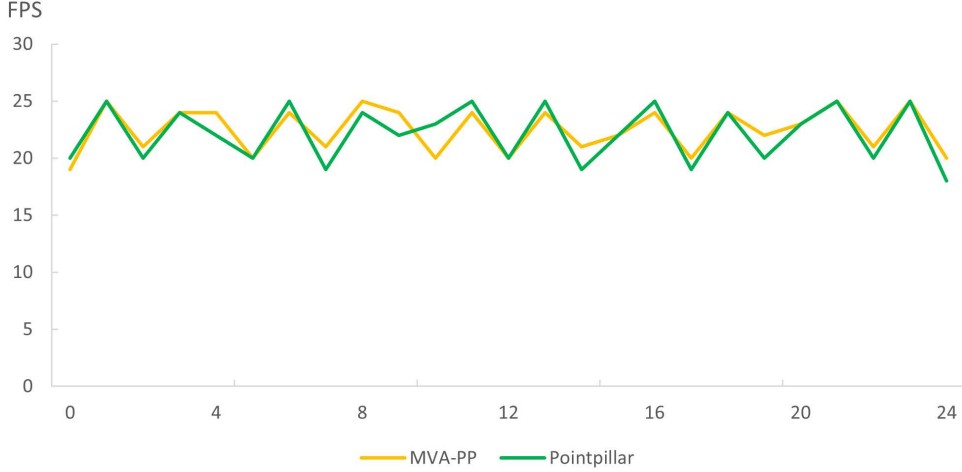

**Fig 10. Line chart of frame rate.**



algorithm, the backbone network is improved, CBAM and CA are introduced to construct the Attention Deconv module, and the first and second stages of upsampling are strengthened to enhance the network's ability to extract large-scale semantic information. Transformer is introduced to construct the Transformer Deconv module, and the upsampling module in the third stage of the backbone network is refined to enhance the ability of network to extract small-scale semantic information and improve the detection accuracy of cyclist class. The algorithm can meet the requirements of an automatic driving system for real-time obstacle detection and has good real-time detection performance.

On the basis of the work in this paper, more efficient point cloud feature extraction modules will be further explored in the future to enhance the ability of point cloud extraction, so as to solve the problem of sparse convolution for fuzzy features and further improve the detection accuracy of targets. The use of cavity convolution is proposed to enlarge the network's receptive field and thus improve the network's detection ability.

## Author contributions

**Conceptualization:** Xinpeng Yao, Jingmei Zhou.

**Data curation:** Zijian Wang.

**Investigation:** Xinpeng Yao, Jingmei Zhou.

**Methodology:** Peiyuan Liu.

**Resources:** Xinpeng Yao, Jingmei Zhou, Zijian Wang, Songhua Fan.

**Software:** Peiyuan Liu.

**Supervision:** Xinpeng Yao.

**Validation:** Yuchen Wang.

**Writing – original draft:** Peiyuan Liu.

**Writing – review & editing:** Peiyuan Liu.

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
