## [Decision Letter · Decision Letter 0]

Dear Dr. Liu,

Thank you for submitting your manuscript to PLOS ONE. After careful consideration, we feel that it has merit but does not fully meet PLOS ONE’s publication criteria as it currently stands. Therefore, we invite you to submit a revised version of the manuscript that addresses the points raised during the review process.

We look forward to receiving your revised manuscript.

Kind regards,

Xin Xu, Ph.D.

Academic Editor

PLOS ONE

Journal Requirements:

This work was supported in part by Open Project of Shandong Key Laboratory of Smart Transportation (Preparation) under Grant 2021SDKLST004, National Natural Science Foundation of China under Grants 52472337 and 52302491, China Postdoctoral Science Foundation under Grant 2023T160129, Key Science and Technology Project of Ministry of Transport under Grant 2022-ZD6-079, Research Funds for the Interdisciplinary Projects, CHU under Grant 300104240911, Natural Science Basic Research Program of Shaanxi under Grant 2023-JC-YB-523, Key Research and Development Program of Shaanxi under Grant 2023-YBGY-119.

Reviewers' comments:

Reviewer's Responses to Questions

**Comments to the Author**

1. Is the manuscript technically sound, and do the data support the conclusions?

Reviewer #1: Yes

Reviewer #2: Yes

2. Has the statistical analysis been performed appropriately and rigorously?

Reviewer #1: Yes

Reviewer #2: Yes

3. Have the authors made all data underlying the findings in their manuscript fully available?

Reviewer #1: Yes

Reviewer #2: Yes

4. Is the manuscript presented in an intelligible fashion and written in standard English?

Reviewer #1: Yes

Reviewer #2: Yes

Reviewer #1: Strength:

1. The MAT-PointPillars proposed in the paper achieves promising 3D average detection accuracy (AP3D) on the KTTI dataset.

2. The logic of the article is clear.

Weakness:

1. Figure 9 is a screenshot.

2. Insufficient number of comparison methods in the article. There are only two comparison methods on the KTTI dataset.

3. The article lacks an analysis of ablation experiments regarding why CBAM achieves higher detection accuracy compared to CA.

Additional Comments For Authors:

1. Is the chart in the article in the wrong place? Why are they all placed at the end of the article?

Reviewer #2: 1. Inconsistent tenses throughout the paper.

2. Unreasonable placement of figures.

3. Experiments are conducted on only one dataset, which lacks credibility.

4. Lack of exploration of the effect of the number of layers on performance in the ablation experiments.

5. Insufficient number of comparison methods.

**Do you want your identity to be public for this peer review?** For information about this choice, including consent withdrawal, please see our Privacy Policy

Reviewer #1: No

Reviewer #2: No

---

## [Author Response · Author response to Decision Letter 1]

3 Mar 2025

Dear Reviewers,

Thank you for your valuable comments on our submitted manuscripts. We are aware that your feedback is essential for enhancing the quality and standard of our work. We have carefully considered and reflected on the various suggestions made by you and have revised the article. The specific responses are as follows:

Reviewer #1 and Reviewer #2:

1. Insufficient number of comparison methods in the article. There are only two comparison methods on the KITTI dataset. / Insufficient number of comparison methods.

Reply: Thank you very much for your suggestion. A more comprehensive comparison is essential to demonstrate the effectiveness of our proposed method. In response to your comment, we have expanded the experimental evaluation by adding two additional methods on the KITTI dataset. Specifically, we included comparisons with PillarNet and SECOND, as shown in Table 4 of the revised manuscript. These new comparisons provide a more robust and thorough analysis of our method's performance relative to existing approaches.

As Table 4 indicates, the proposed MAT-PointPillars algorithm achieves an AP3D_R11(%) of 81.15%, 62.02%, and 58.68% on easy, moderate, and hard samples, which are higher than the benchmark model by 2.44%, 1.19%, and 1.23% respectively. The algorithm obtains 83.85%, 66.38%, and 62.63% of easy, moderate, and hard samples in the BEV, which are higher than the benchmark model 0.81%, 0.87%, and 0.15%, respectively. Compared with VoxelNet, PillarNet, the AP3D_R11(%) and APBEV_R11(%) of MAT-PointPillars are higher than those of VoxelNet and PillarNet in easy, moderate, and hard tasks. Compared with SECOND, AP3D_R11(%) in easy and hard samples are higher than SECOND by 4.81%, 0.18% respectively. Our algorithm mainly focuses on 3D object detection, so AP3D_R11(%) has good indicators. The algorithm achieves superior detection performance.

The updated results and discussions are now included in the revised manuscript. We believe these additions significantly strengthen the paper and offer readers a clearer understanding of the advancements achieved by our proposed method.

2. Is the chart in the article in the wrong place? Why are they all placed at the end of the article? /Unreasonable placement of figures.

Reply: Thank you very much for your suggestion. The position of the chart in the text is important to the reader's reading experience and understanding. Here is a detailed explanation of where the chart is: In PLOS ONE's submission guidelines, the submission requirement for figures is “Do not include figures in the main manuscript file. Each figure must be prepared and submitted as an individual file.” So, in this article, the figures will be placed centrally at the end of the article.

We understand that reviewers may prefer to embed the chart in the text so that readers can more easily view the figure as they read. We fully agree on the advantages of this approach, especially when the number of figures is small, or the figure is closely related to the content of the text. If the reviewer thinks it is more appropriate to embed the figure in the article, we can adjust the position of the figure in the revision, inserting each figure near the position first mentioned in the article. This will further improve the readability and coherence of the article.

Reviewer #1:

1. Figure 9 is a screenshot.

Reply: Thank you very much for your suggestion. We sincerely apologize for the oversight in including Figure 9 as a screenshot. This was unintentional and does not meet the standards expected for publication. We have replaced the screenshot with a high-quality, version of the figure in the paper.

2. The article lacks an analysis of ablation experiments regarding why CBAM achieves higher detection accuracy compared to CA.

Reply: Thank you very much for your suggestion. To explain why CBAM (Convolutional Block Attention Module) is superior to CA (Coordinate Attention) in detection accuracy, it can be analyzed from the following aspects and discussed in detail combined with the results of ablation experiments:

First, CBAM can enhance feature representation in both channel and space through the attention mechanism of these two dimensions. The channel dimension is unchanged, and the space dimension is compressed. This module focuses on meaningful information in the input image.CA focuses on coordinate information and captures spatial position information by decomposing global pooling into pooling operations in the direction of width and height. This module is concerned with the location information of the target. The advantage of CA is that it can better capture the location information of the target, but the attention mechanism in the channel dimension is weak. Furthermore, in 3D object detection tasks, the size and shape of objects vary greatly, and CBAM can better capture these changes through channels and spatial attention mechanisms, thereby improving detection accuracy.

According to the results of ablation experiments, CBAM is superior to CA in detection accuracy, as shown in Table 5. Specifically, CBAM outperformed CA on both AP3D and APBEV. For example, the combination of CBAM+CBAM reached 81.15% on AP3D, while the combination of CA+CA was only 73.47%.

CBAM's attention mechanism makes it superior to CA in multi-scale feature extraction, complex scene robustness and small target detection. Experimental data verified its effectiveness. The relevant content has been revised and supplemented in the paper.

Reviewer #2:

1. Inconsistent tenses throughout the paper.

Reply: Thank you very much for your suggestion. For the problem of inconsistent tenses, we have made the following adjustments in the revised draft:

Algorithm description: unified use of the present tense. For example, change "The algorithm improved the accuracy" to "The algorithm improves the accuracy".

Experiment part: Unified use of present tense. For example, replace "The model achieved an accuracy of 81.15%" with "The model achieves an accuracy of 81.15%".

Conclusion and future work section: Use present or future tense. For example, replace "We proposed future work" with "We propose future work" or "We will propose future work".

In addition to tense issues, we also thoroughly polish the language of the article to ensure that the expression is clear, accurate, and meets the norms of academic writing. We pay special attention to the problems of verb tense, subject-verb agreement, sentence structure, etc., to ensure that the language quality. The relevant content has been revised in the paper.

2. Experiments are conducted on only one dataset, which lacks credibility.

Reply: Thank you very much for your suggestion. We appreciate the reviewer's valuable feedback regarding the need for experiments on multiple datasets to enhance the credibility of our results. The KITTI dataset is widely recognized as a standard benchmark for 3D object detection in autonomous driving scenarios. It provides a comprehensive evaluation framework with multiple difficulty levels (easy, moderate, and hard), which allows for a thorough assessment of the algorithm's performance under various conditions. Many state-of-the-art methods in 3D object detection, including VoxelNet, PillarNet, SECOND and PointPillars, have been extensively evaluated on the KITTI dataset. By comparing our results with these methods, we ensure that our algorithm's performance is competitive and well-validated within the context of this widely accepted benchmark.

The primary focus of this study was to introduce and validate the effectiveness of the multi-scale attention mechanisms and Transformer Encoder in improving the PointPillars algorithm. The KITTI dataset provided a suitable platform for this purpose, as it allowed us to conduct detailed ablation studies and compare our results with existing methods.

In conclusion, while we recognize the importance of evaluating our algorithm on multiple datasets, we believe that the current results on the KITTI dataset provide meaningful insights into the effectiveness of our proposed method. We are committed to extending our experiments to other datasets in future work to further validate the generalization ability of our approach.

3. Lack of exploration of the effect of the number of layers on performance in the ablation experiments.

Reply: Thank you very much for your suggestion. In our experiments, the number of Transformer Encoder layers was set to 2, which is the maximum supported by our current experimental environment (NVIDIA RTX 3060 Ti GPU). Increasing the number of layers would lead to a significant increase in GPU memory usage, exceeding the GPU's memory capacity. As the number of Transformer Encoder layers increases, the computational complexity of the model also increases significantly, leading to longer inference times and negatively impacting real-time performance.

In our experiments, the 2-layer Transformer Encoder already provides high detection accuracy while maintaining real-time performance (average frame rate of 22.63 fps). Increasing the number of layers might further improve accuracy but would significantly degrade real-time performance. Experiments show that the balance between accuracy and efficiency of 2-layer Transformer Encoder is the optimal solution under current hardware conditions. The effect of the number of layers on algorithm performance has been supplemented and added in the paper.

---

## [Decision Letter · Decision Letter 1]

Dear Dr. Liu,

Thank you for submitting your manuscript to PLOS ONE. After careful consideration, we feel that it has merit but does not fully meet PLOS ONE’s publication criteria as it currently stands. Therefore, we invite you to submit a revised version of the manuscript that addresses the points raised during the review process.

We look forward to receiving your revised manuscript.

Kind regards,

Xin Xu, Ph.D.

Academic Editor

PLOS ONE

Reviewers' comments:

Reviewer's Responses to Questions

**Comments to the Author**

Reviewer #2: All comments have been addressed

Reviewer #3: (No Response)

Reviewer #4: All comments have been addressed

2. Is the manuscript technically sound, and do the data support the conclusions?

Reviewer #2: Yes

Reviewer #3: (No Response)

Reviewer #4: Yes

3. Has the statistical analysis been performed appropriately and rigorously?

Reviewer #2: Yes

Reviewer #3: (No Response)

Reviewer #4: Yes

4. Have the authors made all data underlying the findings in their manuscript fully available?

Reviewer #2: Yes

Reviewer #3: (No Response)

Reviewer #4: Yes

5. Is the manuscript presented in an intelligible fashion and written in standard English?

Reviewer #2: Yes

Reviewer #3: (No Response)

Reviewer #4: No

Reviewer #2: Authors have adequately addressed my comments raised in a previous round of review and I feel that this manuscript is now acceptable for publication.

Reviewer #3: 1.Summary of the research

1.1.Summary

This paper proposes the MAT Point Pillars algorithm, which improves the 3D object detection framework PointPillars based on cylindrical objects by integrating multi-scale attention mechanisms and Transformer encoders, optimizing the detection performance for small and irregular objects. In addition, this article constructs a 3D object detection system for real-world scene detection.

1.2.Strengths

(1)This article has a rigorous structure and standardized grammar.

(2)The article introduces a multi-scale attention mechanism to optimize the 3D object detection algorithm, significantly improving its performance.

(3)The article constructs a real-time 3D object detection system, integrates the improved algorithm into practical applications, and verifies the effectiveness of the algorithm in real-world scenarios.

1.3.Weakness

(1)Lack of comparison with existing Transform based 3D object detection algorithms.

(2)The effectiveness of the algorithm was only verified on the KITTI dataset in the article, which cannot prove its generalization ability.

1.4.Recommendation

Minor Revision.

2.Examples and evidence

(1)Can you try to explain in the article why we need to use multi-scale attention mechanisms and what are the benefits of multi-scale attention.

(2)Can you explain why CMBA Attention, CA Attention, and Transform are used in three different scale branches? Have you tried using one attention mechanism or changing the order for all of them.

(3)Please maintain consistency in the font of the text, caption, and main text in the image, such as Figure 4 and Figure 8.

(4)Add citations to the following papers: "Towards Generalizable Person Re-identification with a Bi-stream Generative Model" and "Mix-Modality Person Re-Identification: A New and Practical Paradigm".

Reviewer #4: The paper is somewhat well written and the proposed method is reasonable and seems to be correct. The submitted manuscript is to be revised in the following ways:

1. Explain how the solution guarantees desired system performance?

2. What is the advantage in this article in relation to existing methodologies? Discussions and explanations should be provided on this issue.

3. Comparative experiments are suggested to demonstrate the superiority of the proposed approach. And more discussions should be given to clearly demonstrate the effectiveness of the obtained results.

4. The difficulty encountered in this research should also be discussed in order to show the present research is not a trivial extension of existing methodologies.

5. The literature review of topic considered in the paper should be better structured. It should lead to introducing what the motivation of this paper is, what has not been done before and why the authors addressed and studied this system structure.

6. It is unclear how original the study is in terms of theoretical advancements, further information on this matter is needed. The Introduction section needs to include a few references to emphasize the contribution: Enhanced Feature Extraction YOLO Industrial Small Object Detection Algorithm based on Receptive-Field Attention and Multi-scale Features, Measurement Science and Technology; End-to-end multi-scale residual network with parallel attention mechanism for fault diagnosis under noise and small samples, ISA Transactions; Quantized Iterative Learning Control of Communication Constrained System with the Encoding and Decoding Mechanism, Transactions of the Institute of Measurement and Control; and in that way point out additional modern strategies and opportunities. Please be aware that your text will be more up-to-date if your references are up-to-date.

7. A consistent style for included references should be followed, and every information should be verified. It is crucial to verify all cited papers, fill in any gaps with Volumes, Issues, and Pages, and fix any inaccurate data. In addition, DOI numbers must be provided to any current references that have not yet been appeared in the Volume and Issue.

**Do you want your identity to be public for this peer review?** For information about this choice, including consent withdrawal, please see our Privacy Policy

Reviewer #2: No

Reviewer #3: No

Reviewer #4: No

---

## [Author Response · Author response to Decision Letter 2]

7 May 2025

Dear Reviewers,

Thank you for your valuable comments on our submitted manuscripts. We are aware that your feedback is essential for enhancing the quality and standard of our work. We have carefully considered and reflected on the various suggestions made by you and have revised the article. The specific responses are as follows:

Reviewer #3 �

1. Can you try to explain in the article why we need to use multi-scale attention mechanisms and what are the benefits of multi-scale attention.

Reply: Thank you for your suggestion. Multi-scale attention mechanisms are essential for capturing features at different scales, which is particularly important for detecting objects of varying sizes in 3D point clouds. Different scales use different attention mechanisms, and different scales can obtain different features. Small targets like cyclists often have sparse and irregular point distributions, making them challenging to detect. By incorporating multi-scale attention, the network can focus on both local details and global context, improving detection accuracy.

2. Can you explain why CMBA Attention, CA Attention, and Transform are used in three different scale branches? Have you tried using one attention mechanism or changing the order for all of them.

Reply: Thank you for your suggestion. We refer to TPH-YOLOv5, the small-scale feature extraction module based on Transformer integrates the Transformer Encoder and the upsampling module to enhance the network’s ability to extract small-scale features. CBAM (Convolutional Block Attention Module) and CA (Coordinate Attention) are used in the first and second stages to enhance spatial and channel-wise feature extraction for larger-scale objects. As shown in Table 5, in the ablation experiment, we conduct separate ablation experiments on key submodules such as CBAM and CA. For example, we remove the CBAM and CA modules, respectively, and observe the changes in model performance.

3. Please maintain consistency in the font of the text, caption, and main text in the image, such as Figure 4 and Figure 8.

Reply: Thank you for your suggestion. We acknowledge the inconsistency in fonts and will ensure uniformity across all figures and captions in the revised manuscript. The figures and captions will be standardized to match the main text font.

4. Add citations to the following papers: "Towards Generalizable Person Re-identification with a Bi-stream Generative Model" and "Mix-Modality Person Re-Identification: A New and Practical Paradigm".

Reply: Thank you for your suggestion. We will include citations to the suggested papers in the "Introduction" section to broaden the discussion on multi-modal and generative approaches. The relevant content has been added in the text. Please refer to the highlighted part.

Reviewer #4:

1. Explain how the solution guarantees desired system performance?

Reply: Thank you for your suggestion. The proposed MAT-PointPillars algorithm ensures desired system performance through a combination of multi-scale attention mechanisms and Transformer Encoder integration, which enhance feature extraction and detection accuracy while maintaining real-time processing capabilities. By introducing the Attention Deconv module in the first and second stages of the backbone network, the algorithm improves the extraction of large-scale semantic information, while the Transformer Deconv module in the third stage enhances small-scale feature extraction, particularly for challenging targets like cyclists. As Fig 10 indicates, the system operates at an average frame rate of 22.63 fps, exceeding the sampling frequency of conventional LiDAR(10Hz), thus meeting real-time requirements.

Fig 10. Line chart of frame rate

2. What is the advantage in this article in relation to existing methodologies? Discussions and explanations should be provided on this issue.

Reply: Thank you for your suggestion. The proposed algorithm demonstrates significant advantages over existing methodologies in several key aspects. Firstly, it achieves superior detection accuracy, particularly for small and irregular targets like cyclists, with AP3D improvements of 2.44%, 1.19%, and 1.23% across easy, moderate, and hard difficulty levels compared to the baseline PointPillars model. Additionally, the algorithm maintains real-time performance with an average frame rate of 22.63 fps, surpassing conventional LiDAR sampling frequencies, thus balancing accuracy and efficiency. The ablation experiments further validate the effectiveness of the proposed modules. Overall, MAT-PointPillars addresses the limitations of existing methods by combining multi-scale feature extraction, attention mechanisms, and Transformer-based enhancements, resulting in robust detection capabilities for small targets and real-world applications.

3. Comparative experiments are suggested to demonstrate the superiority of the proposed approach. And more discussions should be given to clearly demonstrate the effectiveness of the obtained results.

Reply: Thank you for your suggestion. To demonstrate the superiority of MAT-PointPillars, comparative experiments were conducted against leading 3D object detection methods, including VoxelNet, PillarNet, SECOND, and the baseline PointPillars. The results show that MAT-PointPillars achieves the highest AP3D scores (81.15%, 62.02%, and 58.68% for easy, moderate, and hard cases, respectively), outperforming PointPillars by 2.44%, 1.19%, and 1.23%, and surpassing other methods in both 3D and BEV detection accuracy. The improvements are particularly notable for small targets like cyclists. These experiments and discussions clearly validate that MAT-PointPillars offers a balanced and superior solution for 3D object detection, especially in challenging scenarios with small or occluded objects.

4. The difficulty encountered in this research should also be discussed in order to show the present research is not a trivial extension of existing methodologies.

Reply: Thank you for your suggestion. The research faces two major challenges. First, detecting small and sparse targets (e.g., cyclists) in point clouds is inherently difficult due to limited geometric features and occlusions. To address this, we integrate vision-inspired Transformer Encoders into the 3D detection framework, enabling the model to capture long-range dependencies and enrich semantic context for sparse point clusters—an approach not previously explored in PointPillars-based methods. Second, maintaining real-time performance while incorporating computationally intensive modules posed a significant hurdle. We optimize the system by carefully balancing model depth and efficiency, deploying it on ROS with a streamlined pipeline that achieved 22.63 fps—faster than conventional LiDAR sampling rates. These solutions highlight the non-trivial contributions of the work, bridging the gap between accuracy and real-time feasibility in 3D object detection.

5. The literature review of topic considered in the paper should be better structured. It should lead to introducing what the motivation of this paper is, what has not been done before and why the authors addressed and studied this system structure.

Reply: Thank you for your suggestion. Existing 3D object detection methods, such as voxel-based (VoxelNet, SECOND) and pillar-based (PointPillars) approaches, struggle with accurately detecting small and irregular targets like cyclists due to sparse point cloud data and limited feature representation. While attention mechanisms and multi-scale feature fusion have been explored in 2D vision (e.g., CBAM, Transformer), their direct application to 3D point clouds remains underexplored, especially in balancing real-time performance. Previous works either lacked global context modeling (e.g., PointPillars) or introduced excessive computational overhead (e.g., PV-RCNN), failing to achieve both high accuracy and efficiency. This paper addresses these gaps by proposing MAT-PointPillars, which integrates multi-scale attention (CBAM/CA) and Transformer Encoders to enhance small-target detection.

6. It is unclear how original the study is in terms of theoretical advancements, further information on this matter is needed. The Introduction section needs to include a few references to emphasize the contribution: Enhanced Feature Extraction YOLO Industrial Small Object Detection Algorithm based on Receptive-Field Attention and Multi-scale Features, Measurement Science and Technology; End-to-end multi-scale residual network with parallel attention mechanism for fault diagnosis under noise and small samples, ISA Transactions; Quantized Iterative Learning Control of Communication Constrained System with the Encoding and Decoding Mechanism, Transactions of the Institute of Measurement and Control; and in that way point out additional modern strategies and opportunities. Please be aware that your text will be more up-to-date if your references are up-to-date.

Reply: Thank you for your suggestion. We will include citations to the suggested papers in the "Introduction" section to broaden the discussion on multi-modal and generative approaches. The relevant content has been added in the text. Please refer to the highlighted part.

7. A consistent style for included references should be followed, and every information should be verified. It is crucial to verify all cited papers, fill in any gaps with Volumes, Issues, and Pages, and fix any inaccurate data. In addition, DOI numbers must be provided to any current references that have not yet been appeared in the Volume and Issue.

Reply: Thank you for your suggestion. The manuscript will ensure that all references adhere to a consistent citation style, every detail will be verified for accuracy. And provide the DOI number for the current reference.

---

## [Decision Letter · Decision Letter 2]

MAT-PointPillars: Enhanced PointPillars Algorithm Based on Multi-scale Attention Mechanisms and Transformer

PONE-D-24-57723R2

Dear Dr. Liu,

We’re pleased to inform you that your manuscript has been judged scientifically suitable for publication and will be formally accepted for publication once it meets all outstanding technical requirements.

Kind regards,

Xin Xu, Ph.D.

Academic Editor

PLOS ONE

Additional Editor Comments (optional):

Reviewers' comments:

Reviewer's Responses to Questions

**Comments to the Author**

Reviewer #3: All comments have been addressed

Reviewer #4: All comments have been addressed

2. Is the manuscript technically sound, and do the data support the conclusions?

Reviewer #3: Yes

Reviewer #4: Yes

3. Has the statistical analysis been performed appropriately and rigorously?

Reviewer #3: Yes

Reviewer #4: Yes

4. Have the authors made all data underlying the findings in their manuscript fully available?

Reviewer #3: Yes

Reviewer #4: Yes

5. Is the manuscript presented in an intelligible fashion and written in standard English?

Reviewer #3: Yes

Reviewer #4: Yes

Reviewer #3: (No Response)

Reviewer #4: The authors gave detailed answers to all of the previous comments and revised the manuscript accordingly. This paper can be accepted now.

**Do you want your identity to be public for this peer review?** For information about this choice, including consent withdrawal, please see our Privacy Policy

Reviewer #3: No

Reviewer #4: No

---

## [Editor Report · Acceptance letter]

PONE-D-24-57723R2

PLOS ONE

Dear Dr. Liu,

I'm pleased to inform you that your manuscript has been deemed suitable for publication in PLOS ONE. Congratulations! Your manuscript is now being handed over to our production team.

Kind regards,

on behalf of

Dr. Xin Xu

Academic Editor

PLOS ONE